# Natural Bioactive Compounds from Food Waste: Toxicity and Safety Concerns

**DOI:** 10.3390/foods10071564

**Published:** 2021-07-06

**Authors:** Ana A. Vilas-Boas, Manuela Pintado, Ana L. S. Oliveira

**Affiliations:** CBQF—Centro de Biotecnologia e Química Fina—Laboratório Associado, Escola Superior de Biotecnologia, Universidade Católica Portuguesa, Rua Arquiteto Lobão Vital 172, 4200-374 Porto, Portugal; avboas@ucp.pt (A.A.V.-B.); mpintado@ucp.pt (M.P.)

**Keywords:** bioactive compounds, food waste, toxicity, nanotechnology, regulation

## Abstract

Although synthetic bioactive compounds are approved in many countries for food applications, they are becoming less and less welcome by consumers. Therefore, there has been an increasing interest in replacing these synthetic compounds by natural bioactive compounds. These natural compounds can be used as food additives to maintain the food quality, food safety and appeal, and as food supplements or nutraceuticals to correct nutritional deficiencies, maintain a suitable intake of nutrients, or to support physiological functions, respectively. Recent studies reveal that numerous food wastes, particularly fruit and vegetables byproducts, are a good source of bioactive compounds that can be extracted and reintroduced into the food chain as natural food additives or in food matrices for obtaining nutraceuticals and functional foods. This review addresses general questions concerning the use of fruit and vegetables byproducts as new sources of natural bioactive compounds that are being addressed to foods as natural additives and supplements. Those bioactive compounds must follow the legal requirements and evaluations to assess the risks for human health and their toxicity must be considered before being launched into the market. To overcome the potential health risk while increasing the biological activity, stability and biodistribution of the supplements’ technological alternatives have been studied such as encapsulation of bioactive compounds into micro or nanoparticles or nanoemulsions. This will allow enhancing the stability and release along the gastrointestinal tract in a controlled manner into the specific tissues. This review summarizes the valorization path that a bioactive compound recovered from an agro-food waste can face from the moment their potentialities are exhibited until it reaches the final consumer and the safety and toxicity challenges, they may overcome.

## 1. Introduction

According to the Food and Agriculture Organization (FAO) one-third of the produced food is wasted [1]. The large amounts of agro-food waste represent a challenge for the food processors, but also an important issue for both environment and international economy since they are one of the causes for landfilling to be no longer sustainable [2,3]. Recent studies showed that agro-food wastes must be considered renewable source of added-value bioactive compounds (BCs) [4,5]. The current linear economy model is based on the one acquired concepts from the industrial revolution, which implied a constant supply of products with a short life span, forcing an ever so increasing production to face the consumers’ needs [6]. This linear approach promotes the underexploitation of natural resources giving rise to a significant environmental and economic crisis. Contrarily, a circular economy encompasses the valorization of the waste allowing the extraction of novel ingredients by returning them to the supply chain, boosting the economy while reducing the environmental impact [3].

The majority of global food losses and waste (FLW) comes from the United States which covers approximately 40% of the whole food supply chain, followed by Central and East Asia and North Africa with 32%, European countries represents 20% while Latin America generates 6% of the FLW worldwide [7], which imply a significant impact on biodiversity, human health, and climate change. However, these FLW might be useful due to their high content of BCs in the production of food additives, functional foods, supplements, and nutraceuticals [5]. Hence, the proper management of this FLW will impact the change for a circular economy model and, a new concept perceived as an efficient option on the long run to transform and increment value to the FLW. BCs recovered from the food supply chain waste using sustainable extracting methodologies, will be used as raw material, or as new products, with health benefits creating industries with added value [6].

Current trends in the food industry and the continuous search for healthy products suggest that consumer’s interest in natural and high-quality foods is increasing [8]. Moreover, the worldwide health crisis created by the COVID-19 redirected the current consumer attitude, perception, and behavioral patterns to reduction of food waste and also regarding the food products consumed [9]. Hence, attention to natural BCs has increased in the past year especially due to the consumers’ awareness regarding the evidence that a healthy and balanced diet has a positive impact on health therefore, consumers worldwide have become more health-conscious [10]. Furthermore, agro-food waste is a rich source of different BCs with content depending on the category of the waste, such as fruit and vegetables, dairy, meat and fish, cereals and roots, tubers and oilseed [3]. The FLW are cheap and renewable from which these BCs can be extracted to yield novel functional food products. Over the past few decades, the diversity of functional foods, supplements, and nutraceuticals in the global market is increasing. The global market of nutraceuticals, food antioxidants, and dietary supplements is expected to reach about $210 billion in 2026 [11]. Where Asia and North America are the main nutraceuticals and dietary supplements consumer markets in the world, whereas Europe focuses on collecting clinical evidence regarding the safety and health benefits of the functional products [12]. Overall, the interest of the food industry for more stable, functional, and user-friendly food additives that can be added to a high diversity of food products is increasing [13]. Because each country has specific laws, the legal aspects of natural BCs recovered from food waste used in a circular economy basis have to be defined. Global regulations and policies for the use of resources in a sustainable way must change from the usual processes (animal feeding, composting and anaerobic digestion) to the current ones (incorporation into the food industries) [6].

The reintroduction of recovered agro-food waste BCs faces many challenges including saftey, biological instability, potential contaminations (pathogens, toxins or pesticides) [14]. Because of that, those ingredients are considered as new foods that must undergo safety assessments likely to change according to different countries, yet limited, legislations concerning to the FLW utilization [15]. The stability and ingredients performance in a food system faces many challenges while designing the appropriate delivery systems for food additives, nutraceuticals, and dietary supplements [16]. The main objective of this review is to compile the journey of the BCs recovered from agro-food wastes until the final consumer, highlighting the safety and toxicity challenges that must be overcome.

## 2. Natural Bioactive Compounds

Bioactive compounds (BCs) definition is not consensual in the literature, however, one of the most well-accepted describes that they are “natural or synthetic compounds with the capacity to interact with one or more components in the living tissues and exerting a wide range of effects [17]. This term is not considered in regulations, however, because they can be part of food supplements, additives, nutraceuticals, functional foods, or novel food, the legal requirements consider them [13]. Furthermore, the boundaries between the food supplements, nutraceuticals, functional foods definitions are not clear and sometimes are confused [18]. Unlike dietary supplements and food additives, there are discrepancies and ambiguities in the definition of what nutraceuticals and functional food are, so a high overlap exists between the terms demonstrating wholesale uncertainty about what they are [19]. A deep analysis of the definitions and the inconsistencies found can be studied in a recent review work of Aronson [20]. Briefly, functional food contains certain substances that may be beneficial to the health in higher concentrations than that in conventional food. These compounds may contribute to enhancing the health benefits, being considered qualitatively more than conventional food [12]. The food additives are ingredients added during food processing for a technological function in order to ameliorate food quality and/or food shelf-life [5]. However, nowadays the interest in incorporating functional and natural food additives has gained significant impulse due to consumers becoming more aware of nutraceuticals and dietary supplements benefits in promoting health [13]. Similarly, the market growth of dietary supplements and nutraceuticals has been driven by consumer interest in health and well-being [21]. Nutraceuticals and food supplements are very similar terms, and they can assume several forms (tablet, capsule, etc.,). From the legal perspective, nutraceuticals represent a special part of dietary supplements because they contain ingredients used for preparing pharmaceuticals, but they do not need to pass through the same tests as pharmaceuticals [12,22].

The interest in BCs for food application in different ways continues to grow, powered by the ongoing research efforts to identify the health properties and potential applications of these substances, mainly extracted from natural sources, and coupled with public interest and consumer demand. They were commercialized as medicines because of their pharmacological assets and were mainly obtained from plants, vegetables, and fungi. The main challenge currently is using the non-edible parts of the natural matrices, such as FLW which still contains a high amount of different BCs with important benefits on human health to use for food industry.

Among the agro-food wastes fruits, vegetables, roots, tubers, and cereals together, have the highest wastage rates. The fruit production is dominated by citrus, watermelons, bananas, apples, grapes, and mangoes, while the most produced vegetables are tomatoes, onions, cucumbers and cabbages, and the roots and tubers are carrots and potatoes [23]. Bioactive compounds isolated include mainly polyphenols, tannins, flavonoids, flavanols, vitamins (A and E), essential minerals, fatty acids, volatiles, anthocyanins, and pigments [24]. By-products from the animal industries represent a good source of bioactive peptides and proteins. Those may include derived products from meat and fish (with side streams bones, tendons, skin, parts of the gastrointestinal tract and other internal organs, and blood) [25]. The dairy industry is another valuable source of proteins and peptides, specially from cheese production [26].

Another alternative presented nowadays for large volumes of produced wastes is the production of energy through the use of biomass or as animal feed (Figure 1). Nonetheless, the wastes are still rich in other valuable compounds such as pigments, sugars, organic acids, flavors, phytochemicals, enzymes, antimicrobial compounds, and fibers that could be extracted in order to use as food additive or supplement with higher added value [27]. Furthermore, the recovery of these BCs is a challenging and important task for their return to industrial chains (commercialization) applying the circular economy concept in order to be employed for the arising trends regarding human demand [3].

Several extraction methodologies have been reported for the recovery of bioactive compounds from agro-food wastes that use organic solvents however, there is a growing need for green and sustainable alternatives. The green extraction technologies that have been the focus are the microwave-assisted extraction (MAE), ultrasound-assisted extractions (UAE), and supercritical fluid extraction (SFE) [4]. Recently the use of natural deep eutectic solvents for the bioactive compounds recovery received great attention as an alternative to the conventional extraction which uses organic solvents that may contribute to toxicity, presenting high volatility and non-renewability [28].

Extensive research in the past few years intends to demonstrate the value of FLWas a resource of compounds with bioactivity or technological potential to introduce new, more sustainable, and natural alternatives to the market. The consumer’s preferences for “clean labels” determines the use of natural compounds to replace synthetic ones, motivating the scientific community to search for new sources of natural alternatives, directing their attention to the utilization of agro-food by-products [5]. One of the most reported potential was the fruit and vegetable waste, since it generates a high amount of residue in the food supply chain and is rich in different types of compounds that can be used for different purposes [3].

### 2.1. Antioxidants

Currently, there are a lot of antioxidant compounds which are already being used in the food industry, mainly as a food additive, and are permitted by regulations, such as ascorbic acid, α-tocopherol, rosemary extract [29]. However, the main challenge nowadays is recovering compounds with the same potential from waste and proving their safety and functionality for introduction in the food system [30]. Any by-product can be considered a valuable resource of new antioxidant food additives, e.g., overly ripe berries, non-compliant fruit, peels, pomace, and seeds. A recent study from Muíno et al. [31] described the use of an olive oil waste extract rich in polyphenols as a potential natural antioxidant applied to lamb meat patties, delaying the lipid and protein oxidation while maintaining acceptable color for a prolonged time, extending the product shelf-life to 3 days. Another recent work from Bitalebi et al. [32] observed that apple peel extract (APE) inhibited protein and lipid oxidation in rainbow trout *Oncorhynchus mykiss* mince during refrigerated storage (4 °C). Furthermore, lipid oxidation was inhibited (96 h at 4 °C) lowering the peroxides and thiobarbituric acid-reactive substances (TBARS) when compared to control.

### 2.2. Preservatives

The most common preservatives in the food industry are nitrates (E240-E259) and nitrites (E249-E250) and they are related to the development of carcinogenic compounds responsible for developing cancer [5]. Therefore, natural antimicrobials that can be added to food are mainly terpenes, peptides, polysaccharides, and phenolic compounds, among others with less expression. The agro-food byproducts are a major source of these compounds with several reports showing the potential antimicrobial activity of different BCs. For instance, olive leaf extract was used to reduce bacteria contamination in shrimp and organic leafy greens [33,34]. Meat products quality and shelf-life was increased [35] and the capacity to delay fish microbiological spoilage (*E. coli, L. monocytogenes* and *S. aureus*) resulted in extending fish shelf-life under retail conditions [36]. Moreover, the antimicrobial capacity of pomegranate peel extract were investigated in chicken products [37,38] and, the extract showed good antimicrobial activity against *S. aureus* and *B. cereus*. In general, addition of pomegranate peel extract to popular chicken and meat products enhanced its shelf-life by 2–3 weeks, during chilling temperature [39].

### 2.3. Anti-Browning

The resulted decrease in fresh produce quality and shelf-life results from enzymatic browning [40] and consequently negatively impacts product color, taste, flavor, and nutritional value. One solution presented to reduce this phenomenon is using antioxidant solutions and two of the most effective and traditionally used solutions are the ascorbic acid, its derivatives, and sulfites [8]. The use of natural compounds with anti-browning properties would be of unquestionable importance for better consumer acceptance of these products. Compounds with a strong antioxidant capacity such as phenolic compounds have been presented as potential inhibitors of oxidative enzymes. This was demonstrated by a recent work developed by Cindy Dias et al. [40] that uses extracts from strawberry tree (leaves and branches) and apple byproducts to inhibit polyphenol oxidase and peroxidase.

### 2.4. Colorants

Most available commercial colorants are synthetics however, a few of them like carotenoids and anthocyanins are already obtained from natural sources [5]. Therefore, the use of FLW as new source of colorants could be a way to shift to more natural additives in the food industry, while still maintaining a cost-effective production. The 16 natural pigments permitted in EU were specified in the Regulation (EC) No. 1333/2008. Good examples of natural color additives are the anthocyanins and carotenoids since in nature they are responsible for the blue, red, purple, orange, and yellow coloration of several species in the plant kingdom. Some examples of anthocyanins sources are winery byproducts, red cabbage, black carrots, purple sweet potatoes, and berries [41]. Blackberry residues are one of the most important sources of natural colorants and nutraceuticals (4.31 mg Cy3GlE/g) [42]. Citrus peels and pomace residues are good sources of carotenoids [43,44]. These compounds can be used for natural colors in products, and beyond that enhancing the shelf-life of food and beverages by preventing pathogens and contaminants or off-flavor formation.

### 2.5. Thickeners

Citrus peels are also rich in pectin. Therefore, pectin extracted from citrus peels could be applied as a gelling agent in bakery, confectionery, and in meat products. Water-insoluble fibers as pectin are also considered a functional food that are used to improve gut microbiota health [29,43].

## 3. Food Waste Bioactive Compounds Regulatory and Legislative Issues

The interest in food industry toward development of functional and nutraceutical products is increasing and the BCs extracted from FLW can be re-used in the human food supply chain as nutraceuticals, functional foods [44], additives, supplements, fortification, and other minor ingredients [14].

One of the challenges related with FLW valorization is the lack of effective policy and is growing the necessity to improve the statutory rules, codes of conduct and guidelines. The utilization of BCs recovered from FLW are still limited to scientific research and patents, because there is still a lack of legislation for FLW utilization [15].

### 3.1. European Union

The general food law (GFL), the European framework regulation on foods, has the purpose to assure the maximum protection level of human health and consumer interest while certifying the correct functioning of the internal European market for both food and feed. Furthermore, the GFL is also the founding regulation for EFSA to provide scientific advice and technical support for all European legislation and policies regarding the food and feed safety matters [45].

In Europe, the European Community (EC) Regulation No. 178/2002, Article 2 and Codex Alimentarius guidelines regulate the use of food waste and by-products as food ingredients or as natural food additives. Therefore, when food by-products are proposed to be used as natural additives and do not match the current regulations, a proper authorization as novel food, EC Regulation No. 258/97 (1997), is required [14]. The Novel Food Regulation (NFR) had a specific regulation (EU) 2015/2283 that deals with foods and food ingredients that were not used before 15 May 1997 for human consumption. Novel foods must undergo a safety assessment and the request must follow the EC Recommendation No. 97/618 [15].

When a new nutritional substance is not included in the ingredients list from 15 May 1997, it is recommended to be included in the Annexes of the Directives on foods for particular nutritional uses, of the Directive on food supplements and of the Regulation on fortified foods and should be submitted to the European Commission, Health and Food Safety Directorate-General, Unit E1, Food information and composition, food waste [46].

In 2009, the European Food Safety Authority (EFSA) published the Guideline entitled “Safety assessment of botanicals and botanical preparations intended for use as ingredients in food supplements”. This guidance focused on the botanical preparations intended for use in food supplements, but also applicable to other uses in the food area. The safety assessment considers the identification and nature of source material that includes part of the plant (seeds, leaves, and other bioproducts), manufacturing process (extraction method, solvents used) to obtain botanical preparations and chemical characterization of constituents. The list does not specify the use of agri-food waste as raw material for the preparation of such nutraceuticals. The botanical preparations should not contain toxic substances making required studies on toxicity and toxicokinetic according to the test methods described by Organization for Economic Cooperation and Development (OECD) or in European Commission Directives 87/432/EEC and 67/548/EC—Annex 5 (Authority & Committee, 2009). The bio-waste producers should confirm the safety of new nutraceutical products [47]. Regulation advices to focus toxicity studies on each specific constituent of the nutraceutical [48].

Bioactive compounds can enter the market as food additives to improve technological and sensory functions [49], as a supplement or being presented as a functional food. The different applications will imply different regulations to follow. The authorization procedure for food additives is laid down in Regulation (EC) No 1331/2008 and the safety is regulated by the EFSA [50]. Based on this data, EFSA evaluates the level below which the intake of the substance can be considered safe—the so-called acceptable daily intake (ADI). Once accepted, the commission makes a proposal for possible authorization of the additive and present it for voting at the Standing Committee on the Food Chain and Animal Health (SCoFCAH) which is preceded by a presentation to the Council and the European Parliament. However, they can still reject the proposal if does not comply with the EU legislation conditions [51]. The European regulatory framework considers that functional foods must match the general food laws [45].

Other legislative aspects must be considered when introducing a new compound in the food industry since the development of “eco-extracts” for food application as an additive for instance is governed by various regulations such as No 1333/2008. Another parameter also regulated (Directive 2009/32/EC) is the solvent used to obtain that compounds which establish maximal solvent residue limits in the final foodstuff [52].

### 3.2. Non EU Contries

In the United States, depending on the product application the ingredients can have the nutritional label and be regulated as food, food supplement, diet constituent, or nutritional supplement which the product is marketed [53,54].

The utilization of food wastes as a food component is limited in the USA, due to consumer health and safety. In this case the Food and Drug Administration (FDA) is the main organism responsible in the federal government to ensure food safety and have enforced the Federal Food, Drug and Cosmetic Act (FFDCA) and the related food safety aspect of Public Health Service Act (PHSA 42 US C). The laws are directed to any generic food safety procedure that may be applicable for any input into food processing. The identification and prevention of physical, biological, or chemical hazards that may represent risks to human health is controled by the Hazard Analysis and Critical Control Point (HACCP) concept. Contrarily to human food, the utilization of food waste for animal and poultry feed, is well-regulated by the FFDCA [14].

The ingredients used for functional foods should meet all the requirements described for conventional food in Federal Food, Drug and Cosmetic Act (1938) (latest amendment in 2018) [55]. On the contrary, food supplements have a statutory definition in the US [56]. They can contain vitamins, minerals, herbs and other botanicals, amino acids, dietary substances for use by humans to supplement the diet by increasing the total dietary intake, metabolites, constituents, extracts, or combinations already reported [57]. All food supplements’ ingredients marketed before the implementation of the dietary supplement health and education act of 1994 (DSHEA) are considered safe and they can be used. However, after this date the ingredients launched are considered novel ingredients and they must be assessed by the FDA. New dietary ingredients must contain only dietary ingredients which are present in the food supply and are not chemically altered and also have history of use and evidence of safety [58].

In Japan, functional foods are considered as a specific category whose approval process is made by the Food with Health Claims (FHC) system. The FHC can be divided in two groups depending on the purpose and function: Foods for Specified Health Uses (FOSHU) and Foods with Nutrient Function Claims (FNFC). The Minister of Consumer Affairs Agency of the Government of Japan is responsible for FOSHU approval. They follow specific requirements: proved efficacy and safety on the human organisms, suitable nutritional profile, existence of analytical methods for quality control and the assurance of compatibility with product specifications by the time of consumption [59].

The food supplements are considered “health food” and they are regulated by the general food laws. All the ingredients are regulated by the Food Sanitation Act (1947) (latest amendment in 2018), while all substances designated as medicine are not considered “health food” and so are regulated by the Pharmaceutical Affairs Act (1960) (latest amendment in 2013) [56].

In China the State Administration for Market Regulation (SAMR) issued guidelines on health food labelling [60]. The main competent authorities for supervision of food additives are the national health commission (NHC), which formulate the food safety standards and registration of new food additives, while the state admnistration for market regulation (SAMR) makes the supervision on production and circulation of food additives, inspection and quarantine of import and export food additives. New food additives are required to be registered with NHC and any intention for production, operation, use or import of a new food additive must request a license and should have information related with name, function category, dose level, application, certificates to prove technical necessity and use effect, quality specification requirements, safety assessment materials, raw materials or sources, chemical structure and physical properties, production techniques, toxicology safety assessment documents, and testing reports on quality specifications [61].

In India, no mentions are made to the use of agro-food wastes, but the Food Safety and Standards Authority of India (FSSAI) is the responsible body that regulates the food safety standards. At the moment the functional foods are divided into eight categories: Health Supplements, Nutraceuticals, Food for Special Dietary Use, Food for Special Medical Purpose, Specialty food containing plant or botanicals, Foods containing Probiotics, Prebiotics, and Novel Foods [62].

## 4. Safety Issues Related with Food Waste Valorization

FLW are a valuable source of BCs, such as phenolic compounds, flavonoids, anthocyanins or carotenoids, pectin, dietary fibers, proteins, and enzymes [15]. Due to this richness in several BCs, food by-products encompass important activities, such as antioxidant, anti-inflammatory, anti-proliferative, antidiabetic, and also antimicrobial and antivirus. The utilization of those compounds recovered from food wastes faces several challenges that can compromise product safety, such as biological instability, potential pathogenic contaminations, high water activity, potential for rapid auto-oxidation, and high levels of active enzymes [14]. The following section describes the most frequent challenges and constrains required to assure food safety once used agricultural by-products. Table 1 describes some examples of the potential contaminants that may persist in the extracted BCs and may represent an issue before considering the reintroduction in the human food chain. The most common contaminants referenced are pesticides, mycotoxins, microbial contaminations, heavy metals, and biogenic amines. All these biological hazards cause severe diseases and therefore, several factors must be taken in consideration when validating if the FLW is suitable for the extraction of valuable components.

During conventional pest management, it is expected that pesticides have been used to safeguard crop production. The plant by-products that have been treated with pesticides could potentially cause troubles as the extracts will no longer be seen as a “natural” product and will be considered as an alternative preparation of the regulated agrochemical [63]. Under the European Union legislation (Article 32, Regulation (EC) No. 396/2005), EFSA is responsible for the annual report of pesticide residue levels analysis in foods in the European market [64]. Contamination by pesticides should be prevented when obtaining the source material, since the smallest pesticides contamination detection may cause problems with the legislation. The selection of solvents (e.g., n-hexane and acetonitrile) for the extraction procedures is of maximum importance because they can selectively solubilize and even concentrate pesticides [65].

Another important contaminant that must be controlled is the mycotoxins. Those are fungal secondary metabolites known to be widely distributed worldwide in many foods and feed stuff [66]. The smallest micro-fungi infestation of the plants can lead to contamination with mycotoxins, persisting in the final products which can compromise their use for the production of high-quality supplements and the safety of their consumption [67]. Among all mycotoxins, aflatoxins (AFs), ochratoxin A (OTA), deoxynivalenol (DON), and zearalenone (ZEN) have gained much attention due to their high frequency and severe health effects in humans and animals [66]. Some of the troubleshooting caused by micotoxins in other organisms like humas and/or animals involves effects like carcinogenic, genotoxic, hepatotoxic, teratogenic, estrogenic, immunosuppressive, nephrotoxic, or neurotoxic [68,69]. There are no regulatory limits established for micotoxins levels in herbal-based food supplements however, the maximum regulatory limits for certain mycotoxins in foods have been set under EU regulation No. 1881/2006 [70]. Only two micotoxins were covered by legislation when used herbs, the aflotoxins and ochratoxin A [71]. There is still required the exposure assessment because there are only a few studies on this topic [72].

Many types of waste material still contain large numbers of microbes that lead to a breakdown of protein resulting in the production of strong odors. In many types of agro-food waste, enzymes are still active which accelerate or intensify the reactions related with spoilage [73]. The amount of available water determines whether a microorganism will grow or survive. Molds, are well adapted to conditions of low moisture while others will produce spores or enter survival state until the moisture is high enough for bacterial action. The contamination of agricultural products or by-products is frequently atributted to foodborne pathogens (e.g., *Salmonella* spp. and *E coli* O157:H7) and the application of contaminated fertilizer especially from animal source is on the basis of that [74].

Metals contaminations (e.g., lead, mercury, arsenic, and cadmium) are particularly disturbing because of their regular presence in food supplements and the inherent toxicological concern that they raise. Factors that are based on metal contamination, in plant-based supplements, are the chemical soil composition, the plant characteristics, and its growing conditions, as well as the lack of purity, extraction techniques, formulation/manufacturing, transport, and storage conditions [75]. One example is the rice bran used as a health food supplement still it contains levels of inorganic arsenic considered carcinogen (∼1 mg/kg dry weight), which are around 10–20 fold higher than the concentrations found in bulk grain [76].

Biogenic amines are basic nitrogenous compounds formed mainly by the decarboxylation of amino acids or by amination and transamination of aldehydes and ketones. They can have unwanted toxicological effects in humans that include rash, edema, headaches, hypotension, vomiting, palpitations, diarrhea, and heart problems [77]. They are found in a broad range of protein-rich foodstuff from either vegetal or animal sources, such as dairy products, meat, fish, and alcoholic beverages and represent markers of food freshness and indicators of inadequate food processing and storage conditions [78].

Safety is a crucial criterion in “eco-extract” definition, while exhibiting no microbial contaminations. To fulfill those requirements, extracts can be pasteurized or submitted to sterilizing filtration at the end of the process [52]. One of the problems related with solvent extraction, is that, depending on the used solvent they can concentrate on the contaminants initially present in raw material. One possibility to avoid this phenomenon is to proceed with plant material decontamination before extraction or during the extraction using absorbent materials [79]. Orange peels extract modified with an hybrid bentonite was employed for the adsorptive removal of carcinogenic mycotoxins [80]. Other materials were also proposed for mycotoxin removal as activated charcoal, hydrated sodium calcium aluminosilicate, and yeast cell wall [81,82,83].

One alternative to avoid contaminants in the extracts, with potential for industrial application, is membrane processes such as microfiltration, ultrafiltration, and nanofiltration. Those can be used to recover, separate and fractionate-specific BCs specially wastewaters, while assuring safety [84]. Integrated processes of microfiltration followed by ultrafiltration have been proposed to recover phenolic compounds such as chlorogenic acid and apigenin-7-O-glucoside from artichoke wastewaters [85], polyphenols from orange press liquor [86], hydroxytyrosol, procatechuic acid, tyrosol, caffeic acid, p-coumaric acid from olive mill wastewaters [87].

The most effective way to control the presence of mycotoxins and guarantee food supplements safety is by prevention of fungal growth and mycotoxin production. For those good agricultural practices (GAP) in field must be implemented, like controlling harvesting and storage conditions improved by technological issues like controlled atmosphere, and also controlling other physical methods like cleaning, milling, etc., [63]. The HACCP implementation is an effective strategy for prevention, control, and periodic monitoring of mycotoxin in all stages from field to the consumer [88].

Some of the strategies (heat treatment or ionizing radiation) used to eliminate bacterial contaminations and to control mycotoxins can compromise the quality of the final product. Some of the mitigation efforts that can be made are the monitorization and prevention of contaminations in an earlier stage of processing or at the raw material itself, preventing their entering into the food processing system [63].

**Table 1 foods-10-01564-t001:** Examples of potential safety hazards related with agro-food by-products valorization.

Contaminants	Type	By-products	Reference
Pesticides	Cyprodinil Dimethomorph Famoxadone	Grape skin extract	[77]
	Dimetoathe Diazinon Fenitrothion Chlorpyrifos Methidathion	Tomato carotenoid extract	[89]
Mycotoxins	Aflatoxin B1 Fumonisin B1 Ochratoxin A (OTA)	Coffee husk and silverskin	[90]
		Orange peel extracts	[80]
		Grape skin extract	[77]
Bacteria’s and molds	Norovirus *Salmonella* *Campylobacter* *Bacillus* *E. coli*	Meat, poultry, dairy, fruits, vegetables, seafood, grains, and nuts	[91]
Heavy metals	Cadmium (Cd) Lead (Pb)	Grape skin extract	[77]
		Green tea extract	[92]
Biogenic amines	Cadaverine Putrescine Ethanolamine Ethylamine	Grape skin extract	[77]
	Phenylethylamine	Rice, soy, almond, coconut and oat press cake	[93]

## 5. Toxicological Evaluation

To protect public health, when a novel food or ingredient is proposed is necessary to ensure consumer health, and for that safety assessments, procedures are proposed in regulations [94]. The toxicological evaluation is fundamental in all safety assessments, whatever the country and the laws that govern them. In EU countries, the toxicological evaluation should be followed by the EFSA guidelines. These guidelines were made to perform the safety assessment of the food additives, vitamins and minerals, novel foods, food supplements, and botanicals and follow the tier toxicity testing approach proposed for food additives in 2012 [95,96]. Whereas in the USA, the evaluation should be followed by FDA rules accordingly the Toxicological Principles for the Safety Assessment of Direct Food Additives and Color Additives Used in Food, commonly known as the Redbook [97]. In EU countries, the EFSA guidelines use a tiered approach, i.e., sequential testing strategy, where it is divided into three groups. The first group, tier 1 includes the minimal data required for all compounds. The following group is the tier 2 and is mandatory for compounds that are absorbed and exhibit in vitro toxicity or genotoxicity in the gastrointestinal system, i.e., compounds that test positive in tier 1 tests. Lastly, the tier 3, which should be accomplished if after tier 2 tests the compounds show bioaccumulation, in vivo genotoxicity, and chronic toxicity. In this tier, a case-by-case approach must be established considering all the available data [18,95]. So, all compounds must be analyzed for the minimum tests required (tier 1) and depending on the results they may need further testing (Figure 2). For example, in vitro positive results in genotoxicity testes (Tier 1) obligate a follow-up for in vivo genotoxicity tests (Tier 2). Moreover, the tiered approach, is designed to evaluate four core areas: toxicokinetics, genotoxicity, toxicity, and reproductive and developmental toxicity [18,95]. Accordingly the guidelines provided by EFSA [95], there are several general issues that should be considered in the design, conduct, and interpretation of toxicological studies for submission to approval, the most important are: (1) The studies should be performed with the additive according to the proposed specifications and should be produced according to the application; (2) studies with animals and humans should comply the EU standards and regulations for ethical approval and welfare standards; (3) studies in animals should follow the internationally agreed test guidelines (Directive 2010/63/EU [98]); (4) toxicokinetics and toxicity of food additives in animals should be conducted using internationally agreed test guidelines described in OECD test guidelines (OECD TG) or in Council Regulation (EC) No. 440/2008 [99]; (5) non-clinical studies should follow the principles of Good Laboratory Practice (GLP) described in Directive 2004/10/EC19 [100]; (6) the oral route should be selected for testing substances and the bioactive ingredients should be added to solid food, or to both solid and liquid, and normally via the diet. For substances applied in beverages, administration via drinking water may be the best option; (7) botanical food additives derived from conventional food sources with a long-term history of food use, may benefit from a “presumption of safety” when an adequate information exists.

Table 2 summarizes the EFSA proposed toxicity tests needed in each tier for each of the core areas. As can be observed, most of the studies should be performed using the methods described in the OECD. These methods are the internationally established test guidelines with the aim to evaluate the effects of chemicals on human health and the environment. The OECD guidelines are split into five sections and, the guidelines are compiled in Section 4. Health effects are the ones used for toxicity tests of the sources of vitamins and minerals, novel foods, and/or botanicals ingredients. Nevertheless, all studies performed should be based on good laboratory practice (GLP) [18,95].

Toxicokinetic provides information about the system exposure to the substance and the process involved in their absorption, distribution, metabolism, and excretion (ADME) [101]. Furthermore, it helps to relate the chemical concentration to the observed toxicity effect and to understand the mode of action of the chemical compound and its metabolites. Additionally, it consents the selection of the appropriate doses for toxicity studies. The main objective of Tier 1 tests is to establish the stability and whether compound or breakdown products are absorbed from the gastrointestinal tract (GIT) through established model’s studies, including gut microbiota (including in vitro, in vivo and ex vivo absorption and bioavailability models). Coecke et al. [102] reported several different methods for the evaluation of the toxicokinetics, including using chamber and inverted sac model. If negligible absorption and breakdown products is confirmed, a restricted number of studies would be accepted. In the case of the compound absorption being small, its metabolites or breakdown products from the GIT study, then Tier 2 should be carried out according to the OECD TG 417, using in vivo studies with rats. The trigger that demands tier 3 studies is the possible bioaccumulation. In this case, toxicokinetic studies with repeated doses in animals will be necessary.

The genotoxicity evaluates the effect of the substances on the DNA, i.e., mutagenicity, and genotoxicity. It is known that genetic modification and germ cells are associated with severe health effects such as degenerative diseases and cancer, even occur at low exposure levels [103]. Furthermore, the alteration on the DNA may induce abortions, infertility, and/or heritable damage. Taken in account the harmful consequences of genetic damage to human health, genotoxicity and mutagenic potential are the basic components of the risk assessment. A basic battery of in vitro tests is required for all the compounds (Tier 1) evaluating induction of gene mutation, structural and numerical chromosomal alteration. For instance, research works of Ribeiro et. al, [104] analyze the genotoxicity of a new ingredient developed from FLW, olive pomace powders, with the objective to be incorporated in the food industry as an additive due to high antioxidant and antimicrobial activity and technological properties. The authors evaluated the genotoxicity accordingly using the Ames test with *Salmonella typhimurium* strain (TA 98) and did not find cell viability problems at any of the concentrations tested, which indicates the absence of mutagenicity. However, these works need to be performed in vitro with mammalian cell micronutrients to validate the judgment of no genotoxicity.

The Ames test (OECD, guideline 471) was also used to prove the non-mutagenicity of three nutraceuticals, already commercialized, that were obtained from grape pomace, vitis vinifera, and apple extract representing natural extracts rich in polyphenols [47]. A stilbene rich extract obtained from gravepine shoots was considered safe as a natural additive after a combined analysis of micronucleous test and comet assay (OECD 474, 489) [105].

Therefore, in the case where all in vitro endpoints are clearly negative, it can be concluded that the compound is not a genotoxic hazard. In the event of positive and/or inconclusive and/contradictory results from the basic battery tests it will be necessary a follow-up with in vivo tests, to assess whether the genotoxic potential observed in vitro is expressed in vivo (Tier 2). There is no Tier 3 for these core areas, however the EFSA could be considered a follow-up of Tier 2 results by carcinogenicity studies and germ cell assay (OECD TG 488). When in vitro and in vivo results are not consistent, then a case-by-case should be analyzed.

The main purpose of doing toxicity tests in a food additive, nutraceutical, or food supplement is to provide more information on treatment-related changes in blood, urine, and clinical biochemistry parameters, histopathological changes in organs and tissue after prolonged exposure to the compounds. To establish the main toxicological profile of the substance, showing the target organs and tissues affected, data from the subchronic study (at least 90 days in rodents) should be submitted to EFSA together with the request for use. In the case of positive effects, chronic toxicity (OECD TG 452) and carcinogenicity tests (OECD TG 451) should be performed or, alternatively, a combined protocol to study both in same experiment (OECD TG 453) (Tier 2). Tier 3 tests have been developed to clarify the classical carcinogenicity bioassay previously performed in Tier 2. Moreover, tier 3 may also include specific tests for neurotoxicity, immunotoxicity, and endocrine effects.

Food compounds showing bioavailability must be assessed in reproductive and developmental toxicity (that is, only if it has positive results in subchronic toxicity in Tier 1). These tests provide information about effects and potency on male and female reproduction system (e.g., fertility, pregnancy, prenatal and postnatal survival, growth, functional and behavioral development of the offspring and reproductive capacity of the offspring) and on prenatal development (e.g., lethality, toxic effects on the embryo and fetus, teratogenicity, and sex ratio). However, conclusions on whether tests are necessary will need to be considered in the light of the toxicity data and toxicokinetics available; if the Tier 1 toxicokinetic tests show that the substance is systemically available, it is necessary to perform Tier 2 testing for reproductive and developmental toxicity is required. Where absorption is insignificant, Tier 2 testing for reproductive and developmental toxicity does not need to be performed. Tier 3 is activated by positive results in Tier 2 studies and might include additional studies for endocrine developmental neurotoxicity (OECD TG 426), that will be studied on a case-by-case basis.

In addition to the four core areas evaluation, the EFSA reported other test may be needed to allow an adequate risk evaluation and establish safety including immunotoxicity, hypersensitivity, food intolerance, etc.

Regarding the guidelines that must be followed in the USA for food additives toxicological evaluation, the FDA first presumes that essential toxicological information is necessary for every food additive. After that, the safety data required are dictated by the concept of concern level (CL). Lastly, the CL is evaluated considering the additive potential human intake and its molecular structure. However, the initial estimation of testing requests can be changed if results suggest a substantial or unpredicted adverse effect. The total results of toxicology studies are then used to calculate the acceptable daily intake (ADI) which is then compared with the estimated daily intake (EDI) values; if the EDI is less than ADI, the food additive is safe under these conditions [106]. The concept of CL (low (I), intermediate (II) or high (III)) is fundamental to the toxicological safety evaluation for the additives and its identification is made based on the diagram represented in Figure 3. The level of expected toxicity of a compound is assigned based on its molecular structure (low (A), intermediate (B), or high (C)) and an estimation of cumulative human exposure [97]. Examples of category A compounds include fats, fatty acids, ketones, esters, monocyclic hydrocarbons, acids, and human metabolites of carbohydrates and lipids. Category B includes inorganic salts of iron, copper, zinc, amino acids, and proteins. Category C includes amides and imines, polycyclic aromatic hydrocarbons, compounds with nitro, N-nitroso, azide, and purine groups. Finally, based on the LOC obtained, there are a list of minimum toxicity studies that are required, to support the safety of the food additive (Table 3) for each of the CL (I, II, and III).

Contrarily to EFSA, the FDA divides the main tests into several categories, but the main test required in each of the categories is the same that are required by EFSA for approval in the EU countries. Likewise, EFSA, genotoxicity tests are performed in vitro and in vivo to consider the key types of genetic alterations and types of DNA damage (mutagenicity and genotoxicity). According to FDA, all compounds must perform the in vitro basic tests battery (OECD TG 471 and 487). However, if the cumulative estimated daily intake of the compound exceeds 50 ppb in the diet an in vivo test is required. In EU this test is only performed when positive results are observed at in vitro basic tests battery. Moreover, in this case EFSA requires two more in vivo tests.

The subchronic studies are applied at all compounds like EFSA, however, in this case they call it “short-term toxicity tests” and evaluate in vivo with rodents for at least 28 days (OECD 407), with a group of animals exposed repetitively to the additive in their diet, while at EFSA the subchronic test for all compounds lasts 90 days (OECD 408). In FDA, it is only tested for 90 days for CL II and III compounds. Besides, in the case of CL III, it is also necessary to carry out long-term toxicity tests with dogs conducted for a minimum of 12 months (OECD TG 452). In these studies, blood and urine samples are evaluated periodically throughout the period and, at the end of the study, detailed necropsies and histopathology are performed on high dose and control groups. Effects related to compounds bioaccumulation in tissues should be evident, allowing for the determination of a “no observable adverse effect level” (NOAEL). In the case of subchronic feeding studies for CL III the results help dose selection for the chronic studies. While, at the previous level data are often used for the ultimate determination of safety.

Tests in the category of chronic and carcinogenicity studies are required only for CL III food additives and are often combined into a single study where the lifetime studies have a duration of approximately 104 weeks in two rodent species. The compounds belonging to CLs II and III demands reproductive and developmental testing and should be performed by exposing male and female rodents (20/gender/group) orally to the additive to evaluate several endpoints. The results would give indication that the tested substance have a potential effect on neonatal morbidity and mortality and on the teratogenic. Lastly, metabolism and pharmacokinetic studies and clinical human studies only need to be carried out if indicated by available data or information.

Briefly, food additives from CL I needs a short-term feeding study in a rodent species, at least 28 days in duration and, a short-term test for genetic mutations potential. CL II requires the previous test for CL I plus testing in a 90-day feeding study in a rodent and a nonrodent species, a multigeneration reproduction study with a developmental toxicity phase and metabolism/pharmacokinetic studies. The additives belonging to CL III require a more extensive testing, in addition to the studies in two rodent species, and a chronic feeding study of at least 1 year in a nonrodent species.

Despite the different regulatory frameworks, the overall risk-assessment procedures and measures are similar.

When considering food supplements some differences in the jurisdictions regarding toxicological data required were registered between the EU and USA. In EU the toxicological data referred to EFSA’s Guidance for submission for food additive evaluations (EFSA Panel on Food Additives and Nutrient Sources added to Food [ANS] 2012). In contrast in USA the FDA considers toxicological tests for various scenarios based on whether the anticipated exposure of the substance exceeds historical consumption. Besides that, it considers clinical trial data, and historical use of botanicals as medicines [58].

**Table 3 foods-10-01564-t003:** Approach suggested by the Food and Drug Administration (FDA) for the toxicity evaluation of food additives.

**Toxicity Tests**	**Concern Level I**	**Concern Level II**	**Concern Level III**
Genetic Toxicity Tests	If the cumulative estimated daily intake of a food ingredient is 50 ppb or less but greater than 0.5 ppb: Bacterial reverse mutation assay (OECD TG471). In vitro mammalian cell micronutrients test (OECD 487) or in vitro mouse lymphoma thymidime kinase+/− gene mutation assay. If the cumulative estimated daily intake exceeds 50 ppb in the diet (150 µg per person per day): The two previous tests + in vivo test for chromosomal damage using mammalian erythrocyte micronucleus (OECD TG 474).		
Short-term toxicity tests with rodents	28 Day oral repeated dose study in Rodents (OECD 407). Screening for neurotoxicity and immunotoxicity.		
Subchronic toxicity studies with nonrodents		Study in rodents at least 90 days of repeated dose (OECD TG 408). Screening for neurotoxicity and immunotoxicity.	
One-Year Toxicity Studies with Non-Rodents			Long-term toxicity tests with non-rodents (usually dogs) should be conducted for a minimum of 12 months (OECD TG 452). Screening for neurotoxicity and immunotoxicity.
Chronic toxicity and carcinogenicity			Chronic toxicity studies in a mammalian species (generally the rat) for 12 months (OECD TG 452) Carcinogenicity for 18–24 months period (OECD TG 451).
Reproduction and Development toxicity studies		Exposing male and female rodents (20/gender/group) orally to the additive (OECD TG 421). Extended One-Generation Reproduction toxicity study (OECD TG 443).	Exposing male and female rodents (20/gender/group) orally to the additive (OECD TG 421). Extended One-Generation Reproduction toxicity study (OECD TG 443).
Metabolism and pharmacokinetic studies		If indicated by available data or information.	
Clinical studies (in Humans)			If indicated by available data or information. There is no requirement for obtaining clinical safety data for food additives. However, could be necessary if indicated by available data.

## 6. Challenges for New “Smart-Foods” for Health

The interest in products free from artificial and synthetic additives is growing as well in making formulations that assure the food bioactive ingredients stability and safely delivered to the target organs and cells. In the recent years several food components and nutraceuticals have been encapsulated by different technologies [107]. One of the most common techniques that guarantees the stabilization of sensitive components, controlled release of core material, and that allows the physical separation of reactive or incompatible ingredients and thereby increasing product shelf-life is through the encapsulation technology [16].

According to the European Commission for legislative and policy purposes is considered a “nanomaterial” if 50% or more of the total particles have one or more external dimensions between 1 and 100 nm [108]. Nanomaterials can occur naturally in food structures at a nanometer range as a result of processing or cooking (e.g., emulsions such as mayonnaise) or they can be engineered [108]. In the last case, the material can be metabolized or excreted by the body such as nanoemulsion or nanoencapsulation of nutrients (e.g., vitamins) or they can persist or be slowly soluble as seen in the case of synthetic amorphous silicon (anti-caking agent), nano-silver (antimicrobial agent), and titanium dioxide (food additive) [109]. There are various nanocarriers sources for food application that may include inorganic (e.g., silver, titanium and silicon dioxide, iron oxide, and zinc oxide), organic components like milk proteins, other animal proteins, plant proteins, polysaccharides, lipids (fats and oils), and biodegradable chemical polymers [107]. Most of the bioactive ingredients including polyphenols, fatty acids, lipophilic vitamins and nutraceuticals, aromas and preservatives are intrinsically hydrophobic and for that water and oil dispersions are prepared [110]. The scheme represented in Figure 4 summarizes the most common encapsulation techniques applied in food industry as well some examples of nanocarriers already tested to deliver bioactive ingredients in food systems.

The techniques that could be applied to encapsulate materials are spray dryer, fluid bed dryer, extrusion, liposome techniques, centrifugal separation, rotational suspension separation, and electrostatic deposition [111]. One of the main targets of nanoparticles technology is to protect BCs from degradation along with the gastrointestinal digestion and cellular metabolism, enabling a controlled release of BCs to the target tissues affected by biological disturbances [112]. Some recent examples of that are molecules like vitamin D (a lipophilic molecule) that was encapsulated into fish oil for higher oral bioavailability [113], ferulic acid prepared with shellac into nanofibers exhibited a colon-targeted sustained release acting as a preventive agent for colon cancer [114]. One of the strategies to increase BCs stability is to increase resistance through stomach acidic conditions and a safe release in the intestine. Isolated lactoferrin from camel milk encapsulated with alginate [115] and catechin with starch-based nanoparticles [116] allowed them a controlled release in the intestine.

**Figure 4 foods-10-01564-f004:**
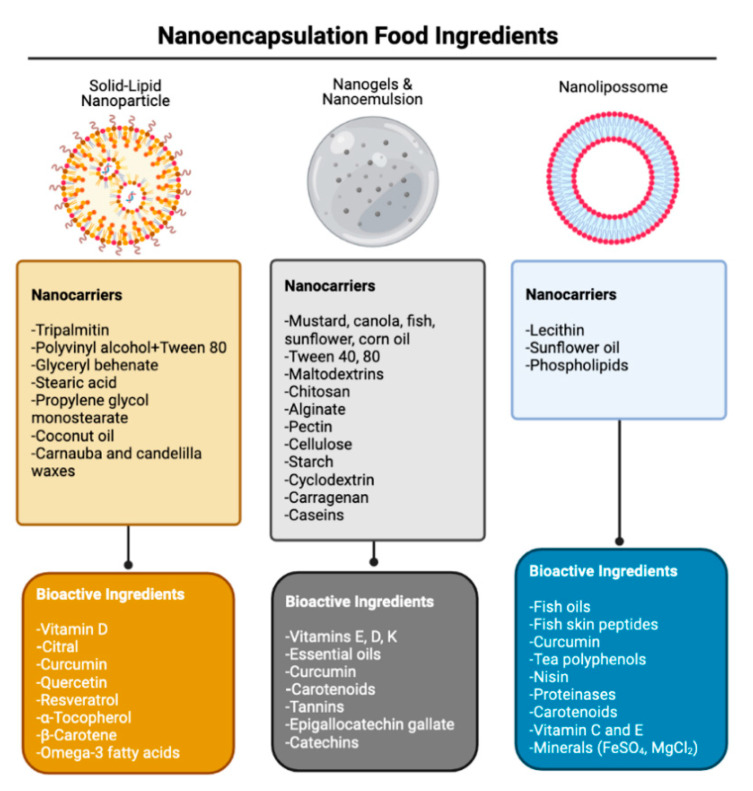
Scheme of the most commonly applied nanocarriers for bioactive compounds [16,107,108,117,118,119,120].

The investment of nanotechnology in food industry market is led by the USA through the National Nanotechnology Initiative (NNI). Other nations that followed them are Japan and the European Union, which have both dedicated considerable resources [121]. Food processing methodologies that involve nanomaterials include integration of nutraceuticals, gelation, and viscosifying agents, nutrient propagation, mineral and vitamin fortification, and nano-encapsulation of flavors [122]. Nanomaterials already make part of commercialized food, but there is still a lack of information related to safety. The Center for Food Safety created a list to alert consumers of common food-related products that contain nanotechnology that includes candies (M&M’s, Skittles), baby bottles, and plastic storage containers [123]. The use of encapsulation for bioactive compounds recovered from agro-food wastes is still restricted to laboratory research and some examples are described in Table 4.

After consumption, the nanomaterials can have different biological destinies in the gastrointestinal system: (1) They are completely digested and absorbed; (2) are partially digested and slightly release the encapsulated compounds; and (3) are resistant to digestion and compounds are thrown out from the digestive system or cross the intestine epithelium and entry to the bloodstream. For the three situations, the toxicological aspect of the components used in nanomaterials production must be considered a potential immunological response as well [129,130]. Once that, the nanoparticles go through the intestinal epithelium without rejection [117]. However, the bioavailability of the encapsulated BCs will increase as particle size decreases, having a direct effect in absorption enhancing the health outcomes [131].

Nanoparticle’s toxicity can be explained across several properties’ dependent on their compositions and structures [117]. Because of their high surface area nanoparticles could adsorb gastrointestinal enzymes altering the normal function of digestion. Depending on the composition, dimensions, morphology, aggregation state, and interfacial properties nanoparticles may accumulate in specific tissues exhibiting toxic effects [132]. Inorganic nanoparticles may generate reactive oxygen species in the cells [133]. The location of bioactive ingredients releases and absorption within the gastrointestinal tract may be altered, hence showing adverse health effects [134]. Toxicity may also be exerted by a higher concentration of the bioactive agents since encapsulation protected them and increase their bioaccessibility [135]. Another harmful effect of nanoparticles is related to their potential interaction with colonic bacteria’s resulting in an alteration of their viability, creating an imbalance in the relative proportions of different bacterial species [136].

Until now there is limited legislation related to the use of nanomaterials in food industry, but agencies and government claim that current legislations ensure the safety of nano-food products [108]. A guidance document entitled “Guidance for the risk assessment of the application of nanoscience and nanotechnologies in the food and feed chain” was prepared in 2011 by European Food Safety Authority (EFSA). These guidelines afford an nanomaterial risks evaluation in food products, where the nanomaterial preparation, amount in final food product, and toxicity are assessed [137].

FDA regulates the nanomaterials included in food products, regarding the safety and regulatory issues in novel food industry technologies, through a guidance for the manufacturers entitled “FDA Regulation of Nanotechnology”. This guidance defines nanomaterials as products with particle size within the nanoscale range (from 1 to 100 nm) and products with physical, chemical, and biological characteristics related to the nanomaterials. Some of the responsibilities are on the manufacturers, which are responsible for monitoring physicochemical properties, impurities, and safety. They are also responsible to submit a regulatory evaluation and indicate a regulatory issue for the consumption of the novel food product [138].

Other countries like Australia, New Zealand, and Korea state that food products with nanomaterials should have a control of the safety requiring experiments to be done before lunching in the food in the market and the related guidelines should also be published [130].

## 7. Conclusions

Natural bioactive compounds represent a special group of health care molecules that could be applied in a great variety of food classes: functional foods, nutraceuticals, dietary supplements, and used as food additives providing to the consumers a natural and sustainable alternative to the synthetic ones. These high value-added compounds could be recovered from food losses and waste but there are many challenges to be addressed in the insertion of these molecules in food additives, nutraceuticals, and dietary supplements, mainly regarding the safety and toxicity. One of the current efforts toward increased ingredient stability, controlled release of the bioactive compounds, and increased product shelf-life is the use of encapsulation technology.

The consciousness of food industry in the importance of the safety aspects, must obey different legal requirements depending of the country where they are applied. The adverse effects of the consumption of natural supplements is relatively rare, and the increasing prevalence of products adulteration with bioactive substances require care from regulators to ensure public safety.

Besides the lack of regulatory, legal provisions, or guidelines for food waste utilization, this review provides a comparison between the European and American regulatory framework as well as the risk assessment criteria that food-related bioactive compounds must follow to ensure product safety. Consumer’s attitude, perception, and behavior related with food waste is changing and their mind-sets are shifting toward “natural” compounds for healthier lifestyle, increasing the responsibility of industries and national authorities in providing a safer product.

## Figures and Tables

**Figure 1 foods-10-01564-f001:**
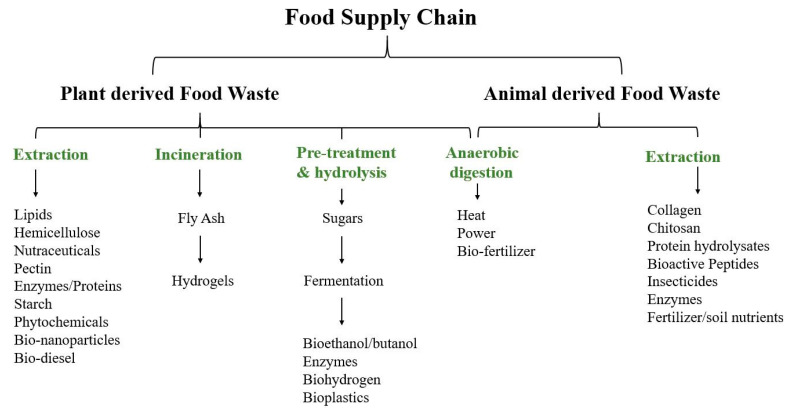
Possible new bioactive compounds extracted from agro-food waste.

**Figure 2 foods-10-01564-f002:**
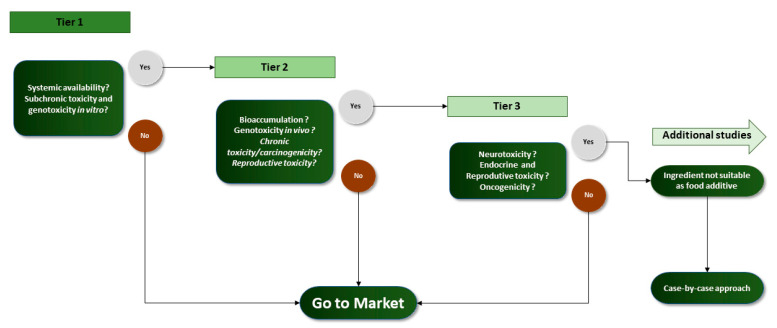
Triggers for the next tier in EFSA approach.

**Figure 3 foods-10-01564-f003:**
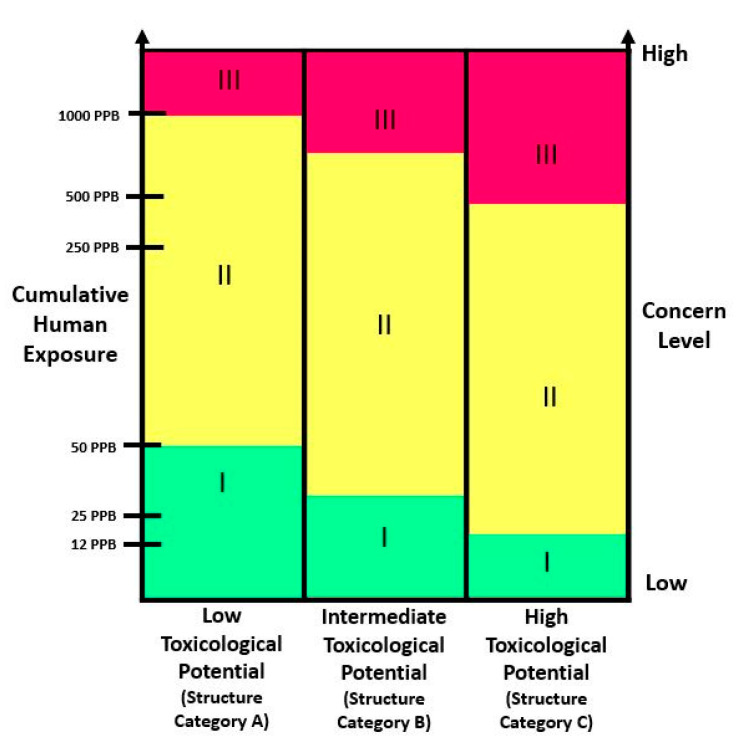
Concern level (CL) defined according to the human exposure and chemical structure. The cumulative human exposure is expressed as ppb of daily dietary consumption of food additive. Adapted from: Redbook [93].

**Table 2 foods-10-01564-t002:** Toxicity tests for food additives evaluation suggested by the European Food Safety Authority (EFSA).

Toxicity Tests	Tier 1	Tier 2	Tier 3
Toxicokinetic	Studies of in vitro gastrointestinal metabolism to establish whether compounds or metabolites are absorbed.	Studies to define ADME and other basic toxicokinetic parameters (T_1/2_, AUC, bioavailability, C_max_ and T_max_) following a single dose (OECD TG 147) together with in vivo assessment of ADME for identification and quantification of metabolites.	Animal studies with repeated administration doses involving studies to steady state which would be approximately five terminal half-lives. Human Clinical Trials.
Genotoxicity	Basic testing battery: Bacterial reverse mutation assay (OECD TG471). In vitro mammalian cell micronutrients test (OECD TG 487).	Follow-up of a positive result in basic test battery. In vivo test for chromosomal damage using mammalian erythrocyte micronucleus (OECD TG 474). (OECD TG 474). In vivo Comet assay (OECD TG 489). Transgenic rodent assay (OECD TG 488).	
Toxicity Tests (subchronic, chronic and carcinogenicity)	Subchronic toxicity study repeated dose in rodents (OECD TG 408, at least 90 days).	Chronic toxicity studies in a mammalian species (generally the rat) for 12 months (OECD TG 452) Carcinogenicity for 18–24 months period (OECD TG 451).	Short-term tests with transgenic mouse models (p53 +/−, rasH2, Tg.AC, Xpa−/− and Xpa−/−p53+/−) (OECD 488) Neurotoxicity, immunotoxicity or endocrine-mediated studies
Reproductive and Developmental toxicity		Prenatal developmental toxicity in rabbit (OECD TG 414). Extended One-Generation Reproduction toxicity study (OECD TG 443). Administration of the test substance should normally be via the diet or by oral gavage to both sexual mature female and male animals covering pre-mating period (at least 2 weeks) and a 2-week mating period.	Studies for endocrine, developmental neurotoxicity (OECD TG 426), and mode of action studies.
Triggers for next tier…	Bioavailability Gastrointestinal toxicity Subchronic toxicity In vitro genotoxicity	Bioaccumulation In vivo genotoxicity Chronic toxicity and carcinogenicity Reproductive & Developmental toxicity	
Additional Studies	Human studies Immunotoxicity Hypersensitivity/allergy Food Intolerance Neurotoxicity		

**Table 4 foods-10-01564-t004:** Nanoencapsulation applied to bioactive compounds recovered from agro-food wastes with the purpose to be reintroduced in food chain.

Food Products	Application	Bioactive Components	Source	Nanoparticles Technique	Reference
Drinking Yogurt	Antioxidant ingredient	Catechin, epicatechin, quercetin, ferulic acid, gallic acid, p-coumaric acid, syringic acid, trans-cinnamic acid, vanillic acid, and vanillin	Cocoa hull waste	Liposomal systems	[124]
Juices and fruit salads	Reduce mycotoxins	p-Coumaric and ferulic acids, epicatechin, quercetin	Grape stem and leaf extracts	Microencapsulation	[125]
Yogurt	Colorant	Betalains	Red pitaya peel	Microencapsulation	[126]
Cupcakes	Antimicrobial	Polyphenols	Pomegranate peel	Microencapsulation	[127]
Beef meatballs	Antimicrobial and antioxidant	Polyphenols (Punicalagin)	Pomegranate peel	Lyophilized pomegranate peel nanoparticles	[128]

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
