# Peer review of "Natural Bioactive Compounds from Food Waste: Toxicity and Safety Concerns"

_foods, 2021, doi:10.3390/foods10071564_

Round 1

Reviewer 1 Report

The topic is very interesting, but I am afraid that the review fails to some degree to fulfil it’  goal since the MS is not well structure and presented.

The language needs extensive revision (some examples are given below in the specific comments), there are many repetitions of the same information and in overall the writing is very dense and talkative, so it is difficult for the reader to follow the meaning in many cases.

As claimed in the introduction, the scope of the MS is t ”to compile the journey of the BCs recovered from agro-food wastes until the final consumer, highlighting the safety and toxicity challenges that must be overcome”. As such I would expect in the section “2. Natural Bioactive Compounds”  to find specific information on the definitions of BCs and their categorization based for example on their applications, chemical structure and benefits their origin/synthesis and benefits. Figure 1 is in my opinion irrelevant. Then the different wastes that could be used as a source for the recovery of the different BCs should be presented and focus should be paid on the methodologies for their recovery. Lines 148-202 provide valuable information, but it is given in a rather scattered way in my opinion which is hard to follow. Illustrations and tables s would be most helpful for organizing this information.

 Moreover, the whole section of toxicological evaluation although being  very interesting is not much to the point. The info is rather generic and do not provide specific information about toxicological tests that are needed or made for specific BCs recovered from wastes. The tables and figures of the section are also too generic. The main point of the review was to evaluate if wastes are appropriate for the exploitation of contained BCs to be used as food additives, so processes, wastes, substances and examples on the evaluation of the above should be included.  

In my opinion whole secctions should be reevaluated and rewritten in a more concise manner and providing more specific information.

Lines 15-16  : please rephrase

line 38  : please rephrase

line 42  : “a short life span” instead of ““a short useful life”

line 44  : Not clear meaning, please rephrase

line 45  : the recovery of what?

line 46  : “reutilization of nutrients and materials” instead of “ingredients reutilization”

,  line 46  : “environmental impact” instead of  “environmental losses”

, lines 52-53: the argument is rather vague. FLW cannot be useful and goals have been set globally for its minimization. What is probably meant is FLW could be exploited instead of wasted

lines 61-64 : Those two sentences are irrelevant to the previous one of lines 60-61 and should better be incorporated somehow in the previous paragraph, or omitted.  

:lines 69-70:  “…a rich source of different BCs with content depending …” instead of “…a rich source of BCs which richness might differ…”

line 71: What is meant by “food waste by-products”? It should be either FW or food production process by-products

line 95 : use only the abbreviation for BC

lines 95-97: the meaning is unclear, please rephrase

Line 101: “…so a high overlap exists…” instead of “…so exist a high overlap…”

Line 102: deep instead of deeply

Lines 104-106 : Please rephrase. A suggestion is “Briefly, functional food contain certain substances that may be beneficial for the health in higher concentration than that in conventional food. “

Line 112: What is meant by “the growth of dietary supplements and nutraceuticals“?

Lines 205-209: Why are you referring to phytochemicals? How are those connected to the information below, (which are actually repetition of the introduction secction)

Line 220: what is meant by arise?

Line 232, 237: Please provide full name when a term appears in the text for the first time

Lines 237-247: Information is out of scope

Lines 248-249 and Lines 249-250:  I cannot understand the connection of the arguments with the above and between them

Lines 251-262: The information here refers to the improvement of levels of different additives and not their origin

Lines 263-266: irrelevant to the topic

Line 336: “The utilization of those compounds in food faces several challenges that can compromise product safety...” does this apply to the BCs that are recovered form food wastes or in general?

Lines 342: Which contaminants do you refer to? Do you refer to pesticides etc that could be present in the waste and recovered together to the BCs?

Lines 372-380: The effect of the presence of microbes on the recovery of BCs is not very clear. The process for BC recovery should include steps that deactivate/eliminate non-deasided microorganisms or their byproducts, shouldn’t they?

Line 44-447: Please rephrase

Line 470: table, not The table

Author Response

Porto, June 24th 2021

Dear Sirs,

Please find attached the answers to the comments made by the reviewer on the manuscript entitled “Natural Bioactive Compounds from Food Waste: Toxicity and Safety Concerns” by Ana A. Vilas-Boas, Manuela E. Pintado and Ana L. S. Oliveira hereby submitted for publication in Foods journal.

The authors would like to thank the suggestion and positive comment given by the reviewers, which certainly contributed to the improvement of our manuscript. All revisions and suggestions were considered, and it is possible to see them in the attached manuscript in highlighted (with track changes activated). Furthermore, an detailed response (in attached) to each reviewer comment were performed.

The authors declare that all the material in this work is original and has not been published. Furthermore, we consider this manuscript has the scientific value and importance to be published in the Foods journal from MDPI following the previous invitation associated with the special issue "Fruit and Vegetable By-Products: Processing, Bioactivities, and Application”

We thank you in advance for your consideration and look forward to hearing from you.

Yours sincerely,

Ana L. S. Oliveira

Reviewer 2 Report

The topic discussed in the manuscript is extremely interesting. The problem of food waste is absolutely topical. Despite the very large proportion of the world's population who suffer from hunger, more and more food is self-wasted. Therefore, the possibility of its further use, due to the presence of natural bioactive ingredients, is important both in sociological, economic and ecological terms.

All elements of the manuscript were prepared correctly. The topic has been described very carefully in every respect. The subsections on legal regulations related to food are very interesting.The reviewer draws attention to minor typing errors appearing in the manuscript.In addition, please very carefully assign the tables and figures that are discussed. In line 341 table 3 is listed and it should be table 1. In line 449 there is figure 1 and it should be 2. Line 470 we have table 4 and it should be 2. Line 561 - there is figure 2 and it should be figure 3. Line 571 - there is table 4, it should be table 3. Line 671 – is figure 1 - should be figure 4. 

Author Response

Porto, June 24th 2021

Dear Sirs,

Please find attached the answers to the reviewer comments made on the manuscript entitled “Natural Bioactive Compounds from Food Waste: Toxicity and Safety Concerns” made by Ana A. Vilas-Boas, Manuela E. Pintado and Ana L. S. Oliveira hereby submitted for publication in Foods journal.

The authors would like to thank the suggestion and positive comment given by the reviewers, which certainly contributed to the improvement of our manuscript. All revisions and suggestions were considered, and it is possible to see them in the attached manuscript in highlighted (with track changes activated). Furthermore, an detailed response (in attached) to each reviewer comment were performed.

The authors declare that all the material in this work is original and has not been published. Furthermore, we consider this manuscript has the scientific value and importance to be published in the Foods journal from MDPI following the previous invitation associated with the special issue "Fruit and Vegetable By-Products: Processing, Bioactivities, and Application”

We thank you in advance for your consideration and look forward to hearing from you.

Yours sincerely,

Ana L. S. Oliveira

Round 2

Reviewer 1 Report

The changes that were made in the MS are adequate

This manuscript is a resubmission of an earlier submission. The following is a list of the peer review reports and author responses from that submission.

Round 1

Reviewer 1 Report

Presented for review manuscript with title: “Natural Bioactive Compounds from Food Waste: Toxicity and Safety Concerns” is very interesting and well written.

Presented manuscript is type Review.

In completely presented sections, Authors show a very important parts connected with topic. Therefore, food waste, bioactive compounds extraction methods are described detail. The problem of food waste have a global range. Authors pointed, that food waste is a great source of a many natural bioactive compounds. Presented Figure 1 give a good picture of extraction methods.

In manuscript text, different chemical and functional bioactive compounds possible extracted from food waste are described in a very logical order. Especially interest from healthy point of view are natural food colorant. It is because of a many synthetic food colorant could illnesses cause among children.

Authors described as well everything what is connect with rules and law food legislation. Together with EU and not-EU law.

It is worth to point important fact presented in Table 1 connected with food hazard as a source and methods extraction are use.

I’m really impress the work made by Authors.

Presented conclusions are in accordance with all points and problems described in whole manuscript.